# Optical conversion of pure spin currents in hybrid molecular devices

May C. Wheeler[1], Fatma Al Ma'Mari [1,2], Matthew Rogers[1], Francisco J. Gonçalves[3], Timothy Moorsom[1], Arne Brataas[4], Robert Stamps[3], Mannan Ali[1], Gavin Burnell [1], B.J. Hickey[1] & Oscar Cespedes [1]

Carbon-based molecules offer unparalleled potential for THz and optical devices controlled by pure spin currents: a low-dissipation flow of electronic spins with no net charge displacement. However, the research so far has been focused on the electrical conversion of the spin imbalance, where molecular materials are used to mimic their crystalline counterparts. Here, we use spin currents to access the molecular dynamics and optical properties of a fullerene layer. The spin mixing conductance across $Py/C_{60}$ interfaces is increased by 10% ($5 \times 10^{18}\,m^{-2}$) under optical irradiation. Measurements show up to a 30% higher light absorbance and a factor of 2 larger photoemission during spin pumping. We also observe a 0.15 THz slowdown and a narrowing of the vibrational peaks. The effects are attributed to changes in the non-radiative damping and energy transfer. This opens new research paths in hybrid magneto-molecular optoelectronics, and the optical detection of spin physics in these materials.

[1] School of Physics and Astronomy, University of Leeds, Leeds LS2 9JT, UK. [2] Department of Physics, Sultan Qaboos University, PO Box 36, Muscat 123, Oman. [3] School of Physics and Astronomy, SUPA, University of Glasgow, Glasgow G12 8QQ, UK. [4] Department of Physics, Norwegian University of Science and Technology, Trondheim, NO 7491, Norway. May C. Wheeler and Fatma Al Ma'Mari contributed equally to this work. Correspondence and requests for materials should be addressed to O.C. (email: o.cespedes@leeds.ac.uk)

Carbon molecules can have extraordinarily long spin lifetimes of up to milliseconds, with applications in organic light emitting diodes (OLEDs), sensors, memories and quantum computing[1–7]. This potential for electronic and optoelectronic applications not notwithstanding, the work on pure spin currents is tightly focused on the electrical signals induced by spin currents via the inverse spin Hall effect (ISHE) and other mechanisms[8–10]. Most commonly, the spin current is generated via spin pumping using ferromagnetic resonance (FMR). There, the power absorbed by a magnet during microwave irradiation is dissipated by the generation of spin waves; collective, propagating magnetic excitations. The angular momentum of the spin waves can be transferred to an adjacent non-magnetic material in the form of a spin current. Spin–orbit coupling (SOC) will transform the spin imbalance into charge separation, leading to a DC voltage transverse to the spin current. Measurements of the ISHE in polymers or carbon-based materials may be done using heavy metal layers, via transport in anisotropic conducting polymers,

using polymers with intra-chain heavy atoms or molecules with an intrinsic SOC[1, 11, 12].

Optical→spin conversion has been demonstrated when illuminating Au nanoparticles embedded in $Pt/BiY_2Fe_5O_{12}$ bilayers[13]. The surface plasmons generated give rise to a pure spin current measured through the ISHE. According to Onsager's principle of microscopic reversibility[14, 15], if surface plasmon resonance can be used to generate a spin current, equally spin pumping could trigger surface plasmons at interfaces with free electrons and where the dielectric permittivity changes sign (see Supplementary Note 1). Given the difference in energy and frequency between magnons and plasmons (meV and GHz vs. eV and THz), the optical effects of this mechanism are likely to be small, although incoherent transfer may play a role in facilitating the conversion[16]. In FMR experiments, the maximum power absorbed by the ferromagnet is typically about 1 $W/m^2$. Even if the spin waves could be efficiently converted, a maximum of ~10 plasmons could be generated per μs and $μm^2$ – taking a typical

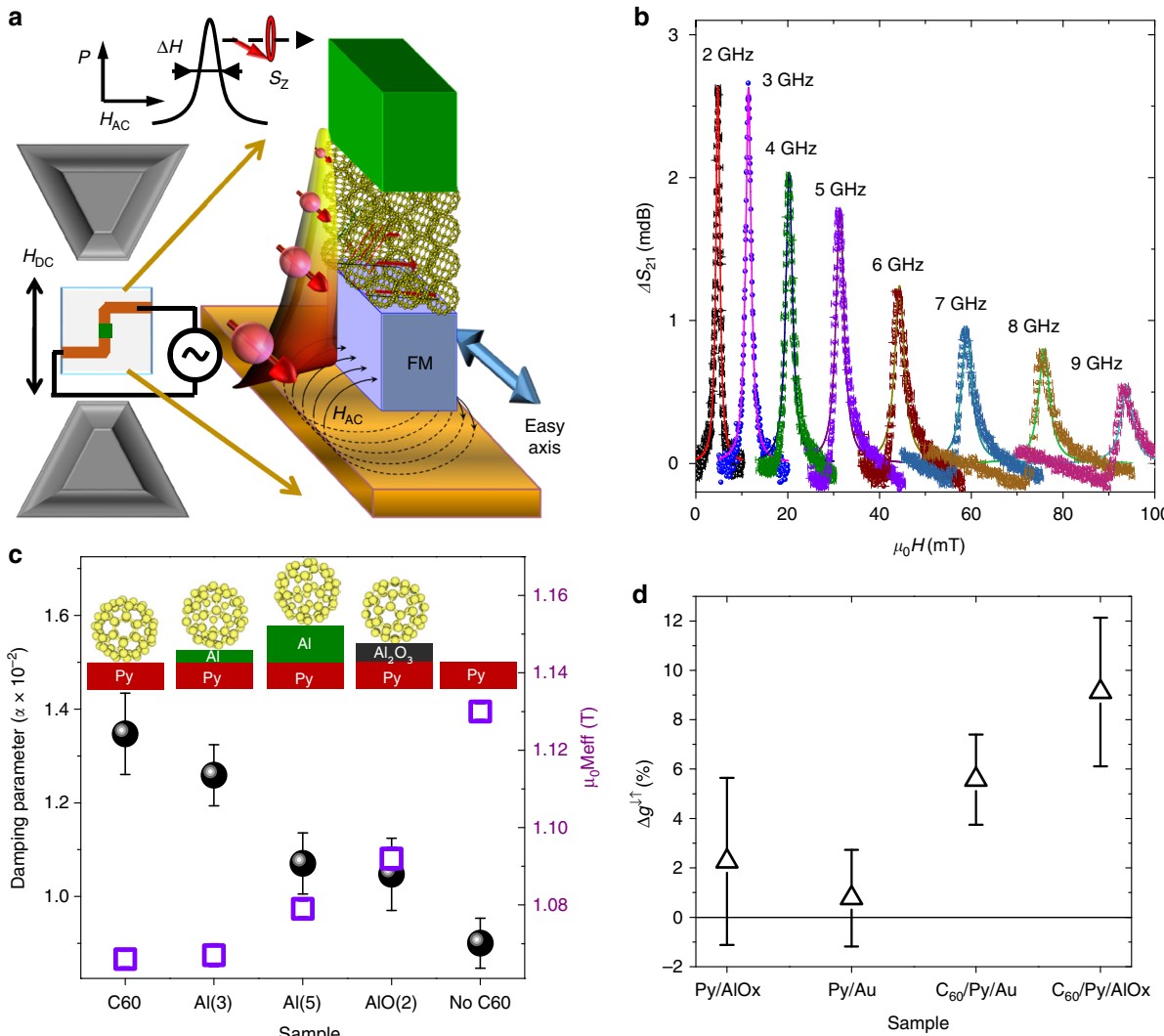

**Fig. 1** FMR spin pumping in $C_{60}$. **a** Principle of operation: the power $P$ absorbed by the ferromagnet FM when the frequency of a microwave field $H_{AC}$ matches the electron spin oscillation $Sz$ in a DC magnetic field $H_{DC}$ is dissipated via spin waves used to manipulate the optical properties of an adjacent $C_{60}$ film. **b**. FMR measured in Py films coated with $C_{60}$. The resonant frequency $\omega_0$ for the power absorbance ($\Delta S_{21}$ parameter) increases with the applied DC field. **c**. Change in effective magnetization $M_{eff}$ (open squares) and Gilbert damping $\alpha$ (dots) for Py with or without $C_{60}$. The presence of an $Al/Al_2O_3$ spacing layer between the Py and $C_{60}$ hinders the magneto-molecular coupling (numbers in brackets are the thickness of the spacer in nm). **d**. Effect of light irradiation in the spin mixing conductance $g^{\downarrow\uparrow}$. A significant enhancement of ~8% is observed in samples with a $C_{60}/Py$ interface. Error bars in **c** and **d** are the mean square error in the Lorentzian fit of the resonance peaks and the Kittel equation

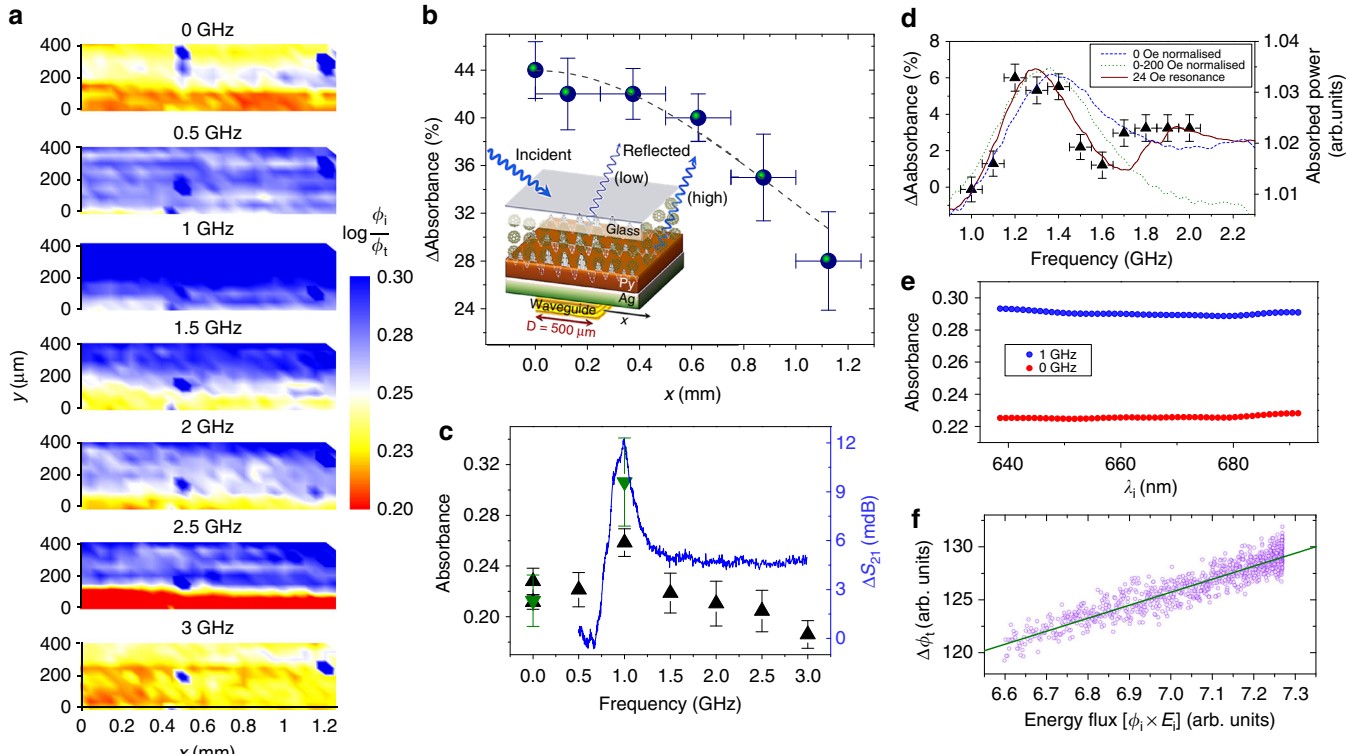

**Fig. 2** Enhanced optical absorbance during spin pumping. The absorbance is defined as the log ratio of the incident ($\phi_i$) and transmitted ($\phi_t$) light intensities for Glass//C$_{60}$(20)/Py(50)/Ag(50) samples – thickness in brackets is in nm. The Py/Ag bilayer acts as a mirror. **a**. Maps at different frequencies show a uniform increase of the absorbance at resonance (8 Oe - 1 GHz); x axis parallel to the waveguide. Error bars in absorbance are the standard error of the mean in 33 measurements, whereas error bars in the x axis are the maximum variation in distance to the waveguide. **b** Absorbance change vs. distance to the waveguide [x]. The effect decays as $(t^2 + x^2)^{-1/2}$, modeled after the AC field generated by the skin current, with x in units of the waveguide width and t a constant (dashed line). Inset, schematic representation of the mechanism for local changes of the optical absorbance via coupling of the electronic structure at the metallo-molecular interface with light-generated plasmons. **c**. FMR power absorption ($\Delta S_{21}$, blue line) and optical absorbance (triangles) vs. frequency. Green and black triangles are two different areas averaged over 99 and 133 points, with error bars the standard error of the mean. **d** Effect reproduced in a sample with the same structure at frequencies close to resonance (10 Oe). Lines show different normalizations for power absorbed at 10 Oe and the shifted resonance at 24 Oe. The asymmetric profile with a local maxima at ~1.9 GHz may be due to sample or waveguide transmission inhomogeneities. Error bars are the standard error of the mean in the absorbance averaged over 51 sample points. **e**. The change in absorbance at 1 GHz irradiation ($\Delta \phi_t$) is proportional to the incident light energy flux: intensity [$\phi_i$] $_{\times}$ photon energy [$E_i$]. **f**. Averaged absorbance vs. incident light wavelength $\lambda_i$ on/off FMR for a spectral region without features (full C$_{60}$ film absorption spectra in Supplementary Fig. 3)

surface plasmon energy of the order of 2–3 eV. However, there may be changes in the interfacial dielectric constant and the coupling between light-generated plasmons and the electronic structure during FMR. It has recently been observed that it is possible to tune the photon–magnon coupling in YIG/Pt structures and observe changes in the corresponding spin pumping[17].

In the following, we shift the emphasis for the spin → optical conversion to the spectroscopic effects of spin currents in molecules. The hyperfine coupling and spin relaxation will play a determinant role in the magneto-optic effects[5, 18–20]. We choose C$_{60}$ as a model system for the spin–optical interaction due to its low hyperfine interaction and long spin diffusion length[6], combined with rich optoelectronic properties that have led to applications in highly efficient solar cells, OLEDs, THz electronic devices and photoassisted magnetization[21–27]. The curvature of the fullerene adds to the SOC, leading to a measurable voltage conversion during spin pumping[12]. The spin current is coupled to the optoelectronic properties of the metallo-molecular interface, and our measurements show a 10% enhancement of the spin current injected into C$_{60}$ during light irradiation. Conversely, the spin current may induce changes to the permittivity, spin triplet density and/or polarizability, leading to a modified Raman

spectrum, higher optical absorption and photoluminescence. Changes in the spin current propagation and optical absorbance are linked to the polarization of the light relative to the quantization axis of the spin current.

## Results

**Spin pumping in C$_{60}$ films**. Our samples include magnetic and C$_{60}$ thin films, where the magnet will act as spin injector when excited at the resonant frequency, and the nanocarbon is the active component for the optical conversion and dynamic effects (Fig. 1a). The resonant frequency at which the spin current is generated depends on the applied DC magnetic field, and is typically 1–20 GHz for fields of 5–3000 Oe using the ferromagnetic alloy Ni$_{80}$Fe$_{20}$ (Py). The linewidth of the resonance, $\Delta H$, increases with the frequency ω as: $\Delta H(\omega) = \Delta H(0) + \alpha\omega/\gamma$, with γ the gyromagnetic ratio and α the Gilbert damping (Fig. 1b and Supplementary Fig. 1)[28]. This damping is intrinsic to the system, and it is correlated to energy dissipation via spin-relaxation[29]. The transfer of angular momentum, and therefore the magnitude of the spin current, can be estimated by comparing α for a Py film with that of the ferromagnet in contact with C$_{60}$[30]. As shown in Fig. 1c, α is up to 50% higher when C$_{60}$ is directly in

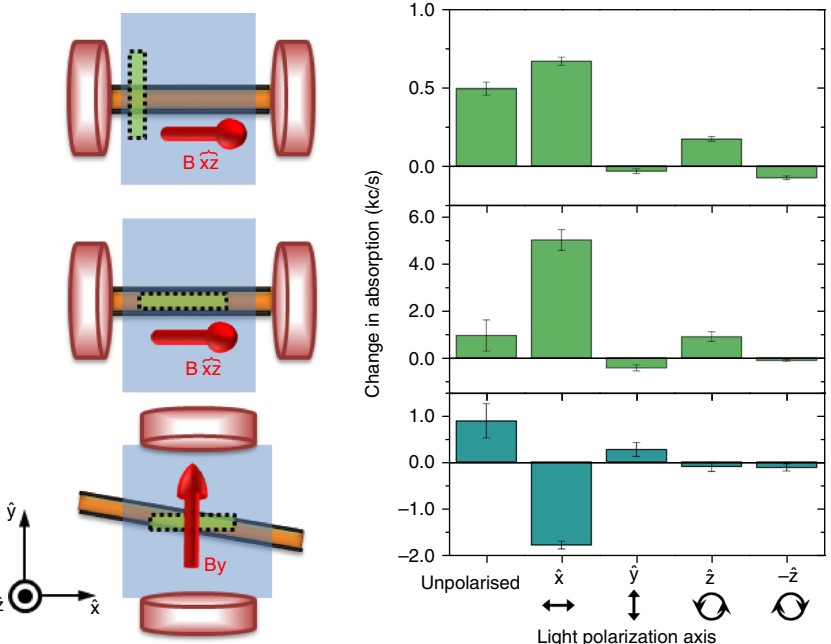

**Fig. 3** Effect of light polarization on optical absorbance. Schematic representations show the magnet (red), waveguide (orange), sample (blue) and measured area (green). top, A field is applied in the $\hat{X}$ direction with an out of plane component (~10% B in the $\hat{z}$ axis). Spin pumping leads to an improved absorbance light polarized in the $\hat{X}$ and $+\hat{Z}$ directions and reduced absorbance for $\hat{Y}$ and $-\hat{Z}$. The effect is maximized when we probe only the film top of the waveguide (middle). bottom, If we alter the sample geometry and apply a magnetic field in the $\hat{Y}$ direction, with the waveguide at an angle, the effect in polarization is reversed, with higher absorbance in the y axis and reduced for the x axis. Change in absorbance for unpolarized light is, as expected, smaller in the top when monitoring in/outside the waveguide and equal in the middle and bottom configurations. Error bars are the standard error of the mean in averages of the optical absorbance over 70 points in a 100 × 300 μm area

contact with Py, evidence that a spin current is injected into the fullerenes. We can moderate the intensity of the spin current by introducing a spacer in between the ferromagnet and the molecules. When a 2 nm $Al_2O_3$ or a 5 nm Al film are inserted at the interface, $\alpha$ is some 20% higher than for Py by itself – caused by the increased distance to the ferromagnet, the absence of free charges to carry the spin information (for $Al_2O_3$) and/or spin mixing at the interfaces and films (Fig. 1c). Changes in the effective magnetization due to the transfer of spin polarized electrons are also reduced on the addition of an Al or $Al_2O_3$ spacer[31, 32], which can be used to manipulate the magnitude of the effects observed (Supplementary Table 1).

The magnitude of the spin current, $J_s$, is proportional to the spin mixing conductance ($g^{\downarrow\uparrow}$), which quantifies the ease of spin pumping into a non-magnetic material[33–35]. At the magnetic resonance, it is defined from the change in the Gilbert damping in the presence of a normal material $\Delta\alpha$, the thickness of the ferromagnet $d_{Py}$, the volume magnetization $4\pi M$, the Landé g-factor $g$, the Bohr magneton ($\mu_B$) and the vacuum permeability ($\mu_0$, equal to 1 and dimensionless in c.g.s. units):

$$g^{\uparrow\downarrow} = \frac{4\pi M d_{Py} \Delta\alpha}{g\mu_B\mu_0} \quad (1)$$

When irradiated with white light (400–900 nm), $g^{\downarrow\uparrow}$ is increased in Py films with $C_{60}$ interfaces by ~10% ($\Delta g^{\uparrow\downarrow}_{\text{light on–light off}} \sim 5 \times 10^{18} \text{ m}^{-2}$)—evidence for a coupling between the optoelectronic properties of the metallo-molecular interface and the propagation of spin currents (Fig. 1d). The change in resistance of the Py film during irradiation is 1%, which would correspond to a temperature change from ~290 to 300 K. This temperature increase will reduce slightly the magnetization M and cause negligible changes in $\alpha$, which would give rise to a

small decrease in the spin mixing conductance[36, 37]. By increasing the source to sample distance, we reduce the light flux by a factor 4–5, leading to an equivalently smaller change in $g^{\downarrow\uparrow}$. We can then introduce a polarizer in between the sample and the source to filter the parallel or perpendicular components of the white light to the magnetic field – and therefore to the direction of the spin quantization axis[30, 38]. We find that light polarized parallel to the field still generates a ~2% enhancement in the spin mixing conductance (~$10^{18} \text{ m}^{-2}$), whereas the perpendicular component of the light causes a reduction in $g^{\downarrow\uparrow}$ at the limit of our sensitivity.

**Optical absorbance and spin pumping effects.** An optical conversion is first apparent in measurements of the optical absorbance of $C_{60}$. Absorbance maps of a 20-nm-thick $C_{60}$ film in contact with Py show a uniformly higher absorbance during spin pumping at the 1 GHz magnetic resonant frequency throughout the probed area of the molecular layer, see Fig. 2a. As the distance to the waveguide generating the microwaves is increased, the spin current decreases back towards its non-excited value with a characteristic length of the order of the width of the waveguide (Fig. 2b). Averaging the absorbance across a large area of the $C_{60}$ film, we can see that changes in the optical absorbance are related to the power absorbance at different frequencies (spin pumping), decaying slightly at high frequencies, this decay being likely due to microwave-induced heating (Fig 2c, d and Supplementary Fig. 2). Even though the absorbance depends only weakly on the incident light wavelength range used in our experiment, we observe that the spin current effect is proportional to the total energy flux of the incident light (Fig. 2e–f). This is probably due to photo-bleaching at high-energy fluxes and independent of the spin current effect–see the Supplementary Figs 1 and 2 for heating effects and the full absorption spectrum.

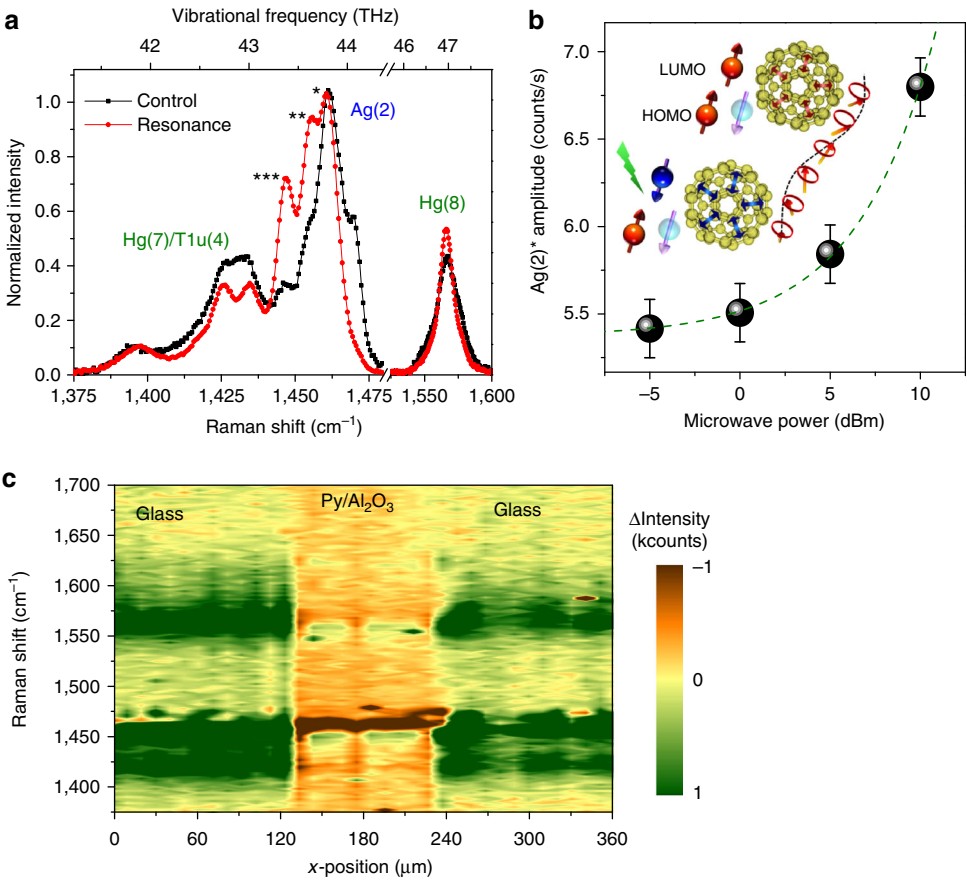

**Fig. 4** Simultaneous Raman-spin pumping measurements. A 1.6 GHz resonance microwave excitation is used in a 20 Oe DC field. **a**. Raman measurements of Glass//$C_{60}$(10)/Py(30) comparing the off (control) and on resonance states of $C_{60}$ on Py. The main Ag(2) peak at 1,467 $cm^{-1}$ splits and shifts to lower frequencies attributed to the formation of triplets and enhanced Coulomb interactions. The splitting in the Hg(7) peak may be due to changes in the molecular symmetry. An in-depth analysis is shown in Fig. S.2. **b**. Amplitude of the Ag(2)* duplicate at different microwave powers. This mode is associated with high spin and charged states promoted by the spin current; schematic as inset (arrows show the atomic movement in the Ag(2) mode). Error bars represent the maximum fluctuation observed in the Raman Ag(2) peak amplitude, and the fit (dashed line) is proportional to the power in Watts. **c**. Change in Raman intensity for maps on/off resonance as a function of wavenumber and position for $C_{60}$ with a 100 μm wide $Al_2O_3$(2)/Py(30) track on top. The Ag and Hg modes for $C_{60}$ under the track are red-shifted during resonance

To improve the efficiency of photovoltaics, the energy output from the enhanced absorbance would need to be larger than the applied FMR power. Taking a typical fullerene-based solar cell with a 6–7% efficiency and 1 kW/$m^2$ irradiance, the 30% increase in optical absorbance while using 10 mW microwave pumping may result in a net energy gain if the excitation is maintained for a coverage $\gtrsim 2$ $cm^2$. Less than 1% of the power applied to the waveguide is absorbed by the ferromagnet (Fig. 2c, d), and other forms of operation could strengthen the effect, for example by integrating the RF current lines lithographed underneath the device. Heating effects result in a reduced optical absorption, and an enhanced magnetoelastic coupling for the ferromagnet used does not result in larger effects, so phonons are unlikely to be the mechanism mediating the interaction. However, in order to improve the optical conversion, we need to disperse any heat generated.

As it was the case for $g^{\downarrow\uparrow}$, there is correlation in the measured absorption changes between the directions of the magnetic field and the electromagnetic (EM) wave vector. Light that is unpolarized or polarized in the direction of the spin quantization axis results in an enhanced optical absorption. However, for light polarized in a perpendicular direction to the spin axis, the optical absorbance is unchanged or even reduced. This reduction with perpendicularly polarized light may be due to a heating effect

reducing the absorbance and not compensated by the spin–optical conversion (Fig. 3).

**Molecular vibrations and Raman spectroscopy**. Molecular vibrations play an important role in the optical and electronic properties of nanocarbon as well, and as such they can be used, for example, in the design of THz $C_{60}$ devices[25, 26]. To measure the fullerene dynamics during spin pumping, we use a FMR-Raman setup – see Supplementary Fig. 2. In Raman spectroscopy, we probe the dynamic polarizability of the molecules, which may be affected by the changes in charge and spin density at the metal interface during spin pumping. The main vibrational peak of $C_{60}$ is the Ag(2) symmetry pentagonal pinch mode at 1,467 $cm^{-1}$, which is highly sensitive to intermolecular interactions. In Fig. 4a, we show the effect of a spin current in a $C_{60}$ film with a 100-μm-wide Py track on top (Glass//$C_{60}$(10)/Py(30)) – thickness in brackets is in nm. Outside this track, the remaining $C_{60}$ acts as a benchmark to ensure the microwaves themselves do not lead to changes in the spectra without a spin current (Glass//$C_{60}$(10)). During spin pumping, the normal Ag(2) mode of $C_{60}$ on Py is quenched, with vibrations transferred to duplicates at 1,445–1,460 $cm^{-1}$. These modes are associated with the spin triplet or magnetized state of $C_{60}$[39, 40], and can be observed

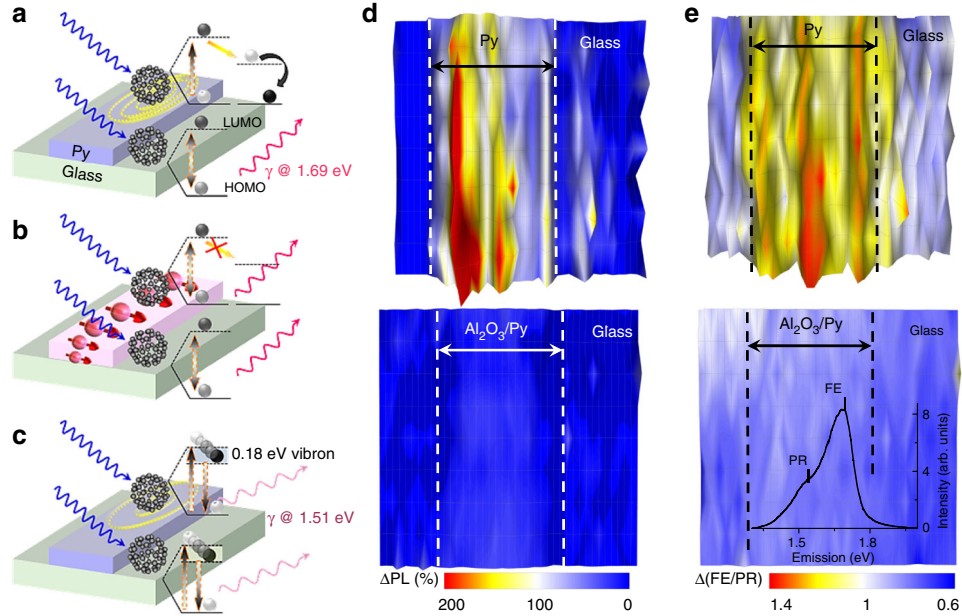

**Fig. 5** PL in $C_{60}$ during spin pumping. Schematics for **a**. light exposure at 2.62 eV generates Frenkel excitons in $C_{60}$ that decay emitting photons at 1.69 eV. For molecules deposited on Py, the molecular dipole transfers energy to the metal leading to non-radiative decay (reduced PL); **b** during spin pumping the dipole-metal energy transfer is less efficient, resulting in an increased PL for $C_{60}$/Py; **c** the radiative decay of the molecular dipole can be promoted via phonon scattering with the Ag(2) vibron, emitting light at 1.51 eV. This process is less affected by the energy transfer or by spin pumping. **d** Photoemission in a 30-nm-thick $C_{60}$ film for an area of $200 \times 40 \mu m^2$. The $C_{60}$ is deposited on a glass substrate and partly covered by a 100 μm wide Py (top figures) or $Al_2O_3$/Py track (bottom). In **d**, we show the PL ratio on/off FMR (1.6 GHz resonance at 20 Oe). Compared to $C_{60}$ on glass, the photoemission intensity on the Py track has increased on average 100% during spin pumping. For the sample with a 2 nm alumina barrier the change is of only 12%. **e**. Ratio of Frenkel exciton to phonon replica photoemission (1.69 vs. 1.52 eV) on/off FMR. This ratio is enhanced by 21% on the Py track during spin pumping. For the Glass/$C_{60}$/$Al_2O_3$/Py sample, no significant changes are observed. Inset shows the PL spectra of the $C_{60}$ films

during surface or tip enhanced spectroscopy[41, 42]. Estimating the molecular population at each vibrational peak, the red shift comparing the vibrational peak for $C_{60}$ on Py off and on resonance during spin pumping translates into an average deceleration of the Ag(2) mode of 0.15 THz. During FMR, the peaks are also observed to narrow, and the intensity of the slower excited modes increases with the microwave power (Fig. 4b and Supplementary Table 2). If we add an $Al_2O_3$ spacer, we suppress the spin current and the vibrational changes are smaller. However, even with a thin insulating barrier there is a significant power damping (see Fig. 1c and Supplementary Figs 4 and 5), and the standard modes are again red-shifted (Fig. 4c).

**Photoluminescence.** $C_{60}$ is fluorescent at ~1.4–1.8 eV via the formation and recombination of polaron-exciton states[43]. Its photoluminescence (PL) is highly dependent on the electronic and magnetic environment[32, 44, 45]. The process is hampered by the symmetry of the molecule that favours non-radiative decay processes, with a PL efficiency of about $10^{-4}$. The PL maximum at 1.69 eV is a result of the direct decay of localized Frenkel excitons. Molecular dynamics lead to lower energy phonon replicas, with the main one at 1.51 eV due to the Ag(2) vibration (1,467 cm$^{-1}$ ~ 0.18 eV; Supplementary Fig. 6). For $C_{60}$ on conducting substrates such as Py, the dynamic molecular dipole transfers energy to the metal via eddy currents, leading to lower PL[40, 45]. During spin pumping, there is a time-varying charge accumulation at the interface that can increase the electrical polarizability and dielectric constant – reducing the energy transfer and increasing the PL (see discussion below). We map this PL change in $C_{60}$ films with a Py track on top and with/without an $Al_2O_3$ barrier. By spin pumping, it is possible to increase the PL by up to $100 \pm 5\%$ for $C_{60}$ directly in contact with

Py. On the other hand, the PL of $C_{60}$ on $C_{60}$/$Al_2O_3$/Py increases just by $12 \pm 3\%$ during resonance (Fig. 5d). The PL of $C_{60}$ away from the Py or $Al_2O_3$/Py track, when the fullerenes are directly on top of a glass substrate, does not change significantly (Fig. 5d and Supplementary Figs 7 and 8). A temperature increase would have the opposite effect: broadening of the Raman peaks and PL quenching with an activation energy of 0.12 eV[46].

The PL spectrum shows as well changes in energy emission. When Py is in contact with $C_{60}$, the emission ratio of the PL due to the Frenkel exciton to this phonon replica is increased by $21 \pm 4\%$ during spin pumping. The effect is limited to a $1 \pm 0.5\%$ reduction during FMR on the addition of an $Al_2O_3$ barrier, see Fig. 5e. This is in agreement with the dynamic shift seen in Fig. 4. The Ag(2) vibrations during spin pumping have lower frequencies corresponding to states in high-spin or charged $C_{60}$ anions -the latter may be due to enhanced charge transfer during spin pumping and/or electron accumulation at the interface. The extinction coefficient, energy transfer to the metal or phonon radiative decay may be reduced so that the effect of spin pumping on the radiative decay is higher for the localized exciton than for the phonon replica.

## Discussion

The optical effects measured are consistent with changes in the spin population and charge density at the metallo-molecular interface. A spin–optical conversion at a molecular interface could be achieved via plasmons and polaritons[47] coupled to the FMR[10, 48]. These surface modes can give rise to a non-resonant enhancement of the quantum yield in the optical absorbance[49–51]. A slowly varying permittivity in the timescale of the microwave excitation will be associated with a large local change in the polarizability modulations responsible for molecular

enhancement of optical absorption. This would be consistent with the observed weak optical wavelength dependence and a strong dependence on the amplitude of fields associated with the microwave resonance.

The enhanced PL for $C_{60}$ on Py during spin pumping can be understood as well in terms of local dielectric changes. For molecules on metallic substrates, the electronic state excited by a photon may decay through non-radiative processes via dipole transfer to the metal (generation of eddy currents in Py) that does not contribute to the PL. The decay rate $\tau^{-1}$ (~$ns^{-1}$) of the molecular electronic state due to this process is proportional to[45, 52]:

$$\tau^{-1} \propto 1 + \frac{1}{d^3}\frac{1}{2k^3}Im\left\{\frac{\varepsilon_m(\omega)-\varepsilon_1}{\varepsilon_m(\omega)+\varepsilon_1}\right\} \qquad (2)$$

where $d$ is the distance from the molecules to the metal layer, $k$ is the wave vector and $\omega$ is the frequency of the EM field generated by the excited state, $\varepsilon_m$ is the complex permittivity of the metal and $\varepsilon_1$ that of the screening layer of charge/spin polarized $C_{60}$ at the interface. A larger $\varepsilon_1$ during spin pumping would lead to a slower decay rate, reducing the non-radiative recombination and increasing the PL. This explains as well the relative enhancement of the Frenkel photo-active decay with respect to the phonon replica, the latter being determined by exciton–phonon coupling and less affected by the energy transfer. The smaller effect of spin pumping for Py/$Al_2O_3$/$C_{60}$ would be due to a reduced spin current and a smaller energy transfer from the molecular dipole to the metal in the presence of an insulating barrier (higher initial PL).

When the fullerenes are deposited on an insulating ferrimagnet (yttrium iron garnet, YIG), small changes on the order of 4% or less are observed in the PL during FMR, with no measurable changes in the Raman spectrum (Supplementary Figs 9 and 10). This can be explained because of the negligible free carrier density and the constant sign of the dielectric constant across the interface, which prevents changes in the electron densities, electrical polarizability or energy transfer to the (now insulating) substrate. A reduced effect with $\Delta PL \sim 6-8\%$ is also measured for $C_{60}$ deposited on thinner magnetic films or Gd-doped Py, where the spin mixing conductance is smaller (Supplementary Table 1). In the case of tunnel barriers, the FMR may lead as well to changes in the DC resistance of the junction (Supplementary Fig. 11).

To maximize the optical conversion and amplification in molecular or inorganic semiconductors, new structures may transform the AC currents originated during spin pumping[53], use pulsed FMR[12] or out of plane DC fields in order to generate surface magnons that may be more efficient in the optical conversion at the interface. The search for optimized material structures could be done by studying the damping coefficient of different magnetic films (e.g. 100% spin polarized half metals) with a range of molecules and film thicknesses on top; larger damping coefficients would imply a better coupling and therefore the potential for stronger effects.

## Methods

**Sample growth.** $C_{60}$ was sublimated in a dual evaporation-sputter deposition system under high vacuum (on the order of $10^{-8}$ Torr) onto a Si/$SiO_2$ or glass substrate. The $C_{60}$ is deposited via high vacuum thermal sublimation and the molecules can also withstand the deposition of metals by sputtering. The $C_{60}$ molecules were from a source of 99.9% purity bought from Sigma-Aldrich. The subsequent metallic layers were sputter deposited in the presence of an in-plane magnetic forming field of 15 mT which induces macroscopic uniaxial anisotropy in a defined direction in the magnetic layers. The metallic materials were deposited at ambient temperature and at a working gas (Ar) pressure of 3.2 mbar. An aluminum cap of 3 nm was deposited on top of the devices to prevent oxidation of the sample. $Al_2O_3$ barriers were formed by plasma oxidation of aluminum films or tracks. Thin

$Al_2O_3$ layers have shown to be effective on magnetically decoupling the molecular layer from the $3d$ metal. The metallic and insulating tracks were deposited through a shadow mask.

**FMR and spectroscopy.** A custom built vector network analyzer-FMR system that can be used simultaneously with Raman spectroscopy, optical transmitivity or photoluminescence measurements was used. The microwaves were applied through a printed circuit board waveguide sitting in a DC magnetic field. White light (400–900 nm) was generated using a 150 W Xe lamp and transmitted via an optical fiber. The samples were positioned face down on the waveguide with the easy-axis of the ferromagnet (defined by the forming field) positioned parallel to the DC magnetic field and the microwaves perpendicular to it. Figure 1b shows the FMR peaks for a Py/$C_{60}$ system as a function of the effective external DC magnetic field, $H_{eff}$. One can extract the effective magnetization, $M_{eff}$, for Py from the frequency of the resonant peak $\omega$ and the applied DC field $H_{eff}$: $\omega = \frac{\gamma}{2\pi}\sqrt{H_{eff}\left(H_{eff}+\mu_0 M_{eff}\right)}$. The samples were deposited on a glass substrate, this is so that it is possible to optically probe and/or irradiate the $C_{60}$ while assuring the magnet is close to the waveguide and excited to resonance. Measurements during irradiation are performed using a Xe lamp and an optic fiber focused on the sample. The absorption, photoluminescence and Raman measurements were carried out using a Horiba-Jobin-Yvon LabRam HR800 system using either a blue (473 nm) or green laser (532 nm) for Raman/luminescence, an incandescent lamp for absorption measurements, and gratings of 1,800 and 600 ln/mm for single spectra and map measurements, respectively.

**Data availability**. The experimental results that support the findings of this study are available in the Research Data Leeds repository (DOI 10.5518/228)[54].

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

## Acknowledgements

We thank Thomas Rylands for the deposition of RF sputtered YIG films and Prof. Gerrit Bauer for discussions on the optical conversion. This work was supported ty EPSRC through the grants EP/K00512X/1, EP/M000923/1 and EP/K036408/1. R.L.S. and F.G. acknowledge support through EPSRC grant EP/M024423/1. F.A.M.M. acknowledges support from Sultan Qaboos University.

## Author contributions

M.C.W. built the microwave-optical setup, conducted experiments and contributed to the data analysis. F.A.M.M. performed the optical absorption under ferromagnetic resonance and FMR with optical irradiation. M.R. carried out measurements of ferromagnetic resonance under optical irradiation. F.J.G. contributed to the measurement of ferromagnetic resonance and the design of the microwave system. A.B. and R.S. contributed to the explanation of the effect. T.M., M.R., M.A., G.B. and B.J.H. optimized the sample growth and setup. O.C. designed the study, analyzed data and wrote the manuscript. All authors discussed the results and commented on the manuscript.

## Additional information

**Competing interests:** The authors declare no competing financial interests.

