## [Peer Review File · Nature Communications]

Reviewer #1 (Remarks to the Author):

Referee report for the article "Optical Conversion of Pure Spin Currents in Hybrid Molecular Devices" by M. C. Wheeler et al.

The manuscript of Wheeler and coworkers explores the very intriguing and yet unexplored idea of converting a spin signal into an optical signal in organic molecules.

Up to now, efforts have been concentrated on the effects induced by spin currents on electric signals, mainly mediated by spin-orbit coupling.

Considering spin to optical conversion means basically to study the influence of magnons, - the fundamental spin excitations - into collective electronic excitations, the plasmons. Since these two fundamental excitations have intrinsically different frequencies, a direct demonstration of their interaction is very difficult.

For this reason, the authors concentrate on the effects of spin currents on the electronic structure of C60 molecules (more specifically on the slight absorbance, on the photoluminescence and on the vibrational modes), that should be mediated by the excited plasmons.

The manuscript is well written, the data are conclusive and their analysis has been carefully performed. Considering the quality of the manuscript as well as its novelty and importance of the topic of the manuscript, I think that it is suitable for publication in Nature Communications.

In the following, I have a few points that the authors should consider before acceptance.

1) The authors observe an increase of 50% in the Gilbert damping when C60 is in direct contact with Py, and explain this with the injection of a spin current in C60, which I find very plausible. On the other hand, they observe an increase of 20% Gilbert damping when a AlOx layer is inserted between Py and C60, and explain this with the presence of pinholes.

I do not doubt that pinholes will be present, however I see no necessity to have a metallic contact to inject a pure spin current. If my thesis is true, than the observation of a reduced Gilbert damping could be more easily explained by considering that the C60 layer is now farer away from the Py surface, and thus the efficiency of spin current injection should be reduced. Do the authors agree on this explanation?

2) The inset in Fig 2 can not be recognized: it is simply too small.

3) The authors discuss the influence of the injected spin current on the Ag(2) mode of C60. Although I agree that Figure 3 a shows that there is an influence on the Ag(2) mode, I think that the data is not fully conclusive on this specific point. To improve this, the authors should also show how the Raman spectrum looks like outside FMR resonance for the C60/Py system, in order to exclude that the observed changes are not simply caused by the presence of the Py layer on to of C60. It could also be that I am misinterpreting Fig 2 a and Fig S2: what do the authors mean with "control" measurement? Is it a Raman spectrum for the C6/Py system off FMR resonance, or is it is the Raman spectrum of the Glas/C60 "control" sample (which was my guess)? Please clarify.

4) Regarding the interpretation of the data. The authors always talk about spin and charge accumulation at the metal-molecule interface. Again, if a pure spin current is generated by spin pumping, then there would be no net charge current flowing, and thus probably also no charge accumulation?

5) Again on the data interpretation. If the enhanced PL is really due to spin accumulation at the metal-organic interface, this would mean that the whole PL signal comes from the interface. Is this really the case? Is the method sensitive enough to detect such a small signal? vWhat about all the rest of the C60 molecules that are not in contact with Py? How thick are the C60 films?

6) Sample preparation: how did you check that C60 are still intact after Py is deposited on top? How about intercalation of Py inside the C60 films? How well defined are the interfaces in the used devices?

Reviewer #2 (Remarks to the Author):

This paper reports modulation of optical properties of C₆₀ (fullerene) put on a ferromagnet (Py) induced by microwave irradiation. They observed modulation of optical absorbance, vibration Raman scattering peaks (shape, position, and intensity), and photo luminescence intensity. Spin current experiments have relied on the inverse spin Hall effect for long years, and discovery of another new method will become an important piece of information for spintronics researchers. However, the paper does not show that the observed signals are relevant to spin current, although the title is "Optical conversion of pure spin current..." The data just shows the spectral modulation at the FMR condition. No typical spin-related phenomenon, such as circular dichroism, polarization rotation, oscillatory angular dependence, or comparison with theories is shown. Indeed, one of the effects of FMR is spin pumping, but it is not all of FMR; there are some other effects, such as local heating, mechanical oscillation etc. Local inhomogeneous heating may induce strain, for instance, which cannot be blocked by Al₂O₃.

Therefore, I regret to conclude that the "spin current induced" is just speculation, and the paper does not meet the criteria of Nature communication.

(minor points)

(a) Figure 2 b caption: What does the function $(t^2+x^2)^{-3/2}$ mean? Is it based on a physical model?

(b) Page 9 line 9: However, the energy transfer may be screened by the charge and spin accumulation: Please elaborate the mechanism.

We thank the referees for their comments and suggestions, which we think have significantly improved our paper. Following their reports, we have performed new measurements and modified the manuscript. The main changes are as follow:

- We have performed measurements of the spin mixing conductance and observed an enhancement with light irradiation for C60/Py interfaces. This effect disappears if the light is polarised perpendicular to the spin quantization axis (page 4, figure 1).
- We have correlated changes in the optical absorbance induced by spin pumping with the direction of the magnetic field and the light polarisation vector (new figure 2).
- We have detailed the explanation for the enhanced photoluminescence during spin pumping, basing our argument on changes to the polarizability/dielectric properties of the interface and the analytic expression for the decay rate developed by Persson and Lang (refs. 45,52)

Other changes and a detailed point-by-point response are below.

Reviewers' comments in blue font and ours in black.

Reviewer #1 (Remarks to the Author):

The manuscript of Wheeler and coworkers explores the very intriguing and yet unexplored idea of converting a spin signal into an optical signal in organic molecules.

Up to now, efforts have been concentrated on the effects induced by spin currents on electric signals, mainly mediated by spin-orbit coupling.

Considering spin to optical conversion means basically to study the influence of magnons, - the fundamental spin excitations - into collective electronic excitations, the plasmons. Since these two fundamental excitations have intrinsically different frequencies, a direct demonstration of their interaction is very difficult.

For this reason, the authors concentrate on the effects of spin currents on the electronic structure of C60 molecules (more specifically on the optical absorbance, on the photoluminescence and on the vibrational modes), that should be mediated by the excited plasmons.

The manuscript is well written, the data are conclusive and their analysis has been carefully performed. Considering the quality of the manuscript as well as its novelty and importance of the topic of the manuscript, I think that it is suitable for publication in Nature Communications.

In the following, I have a few points that the authors should consider before acceptance.

1) The authors observe an increase of 50% in the Gilbert damping when C60 is in direct contact with Py, and explain this with the injection of a spin current in C60, which I find very plausible. On the other hand, they observe an increase of 20% Gilbert damping when a AlOx layer is inserted between Py and C60, and explain this with the presence of pinholes.

I do not doubt that pinholes will be present, however I see no necessity to have a metallic contact to inject a pure spin current. If my thesis is true, than the observation of a reduced Gilbert damping could be more easily explained by considering that the C60 layer is now farer away from the Py surface, and thus the efficiency of spin current injection should be reduced. Do the authors agree on this explanation?

We thank the referee for her/his suggestions and comments. The propagation of spin currents across insulators in the absence of free charges has been reported (e.g. YIG/diamond), but it is not yet fully explained –although it has been suggested that can be due to exchange interactions. Therefore, the referee is correct that a reduced Gilbert damping could be due to the C60 being further from the ferromagnet and we have modified our argument in the paper to take into consideration this possibility. Please note as well our new measurements showing an enhancement in the spin mixing conductance of Py/C60 interfaces with light irradiation (page 4 and figure 1).

2) The inset in Fig 2 can not be recognized: it is simply too small.

We have increased the size of the inset within the figure, and that of the figure itself. The labels are now of equal size for the graph and the inset. If the referee thinks that it still needs to be bigger, we could perhaps move it to the SI or on a separate figure.

3) The authors discuss the influence of the injected spin current on the Ag(2) mode of C60. Although I agree that Figure 3 a shows that there is an influence on the Ag(2) mode, I think that the data is not fully conclusive on this specific point. To improve this, the authors should also show how the Raman spectrum looks like outside FMR resonance for the C60/Py system, in order to exclude that the observed changes are not simply caused by the presence of the Py layer on top of C60. It could also be that I am misinterpreting Fig 2 a and Fig S2: what do the authors mean with “control” measurement? Is it a Raman spectrum for the C6/Py system off FMR resonance, or is it the Raman spectrum of the Glas/C60 “control” sample (which was my guess)? Please clarify.

In the previous version of the manuscript we used the word “control” in two different contexts, causing the confusion. As the referee points out, the Raman spectrum of C60 on Py is different from that of C60 on glass due to charge transfer etc., and cannot be used to establish the effect of spin pumping.

Our comparison is between the Raman spectra (or luminescence etc.) of C60 on the same material but on/off FMR; e.g. Figure 4a shows the Raman spectrum of C60 grown adjacent to Py and measured in the conditions of on/off resonance.

In the previous version of the manuscript, we used “control” to refer as well to the fact that changes during FMR should not be expected for C60 on glass (i.e. the effect is not due to microwaves, but to the spin pumping from the ferromagnet). For this purpose, Figure 4c shows the change in the Raman spectrum during resonance compared to off FMR for C60 on glass (no change in the spectral shape) and on Py (as described). We have made this clear in the new manuscript.

4) Regarding the interpretation of the data. The authors always talk about spin and charge accumulation at the metal-molecule interface. Again, if a pure spin current is generated by spin pumping, then there would be no net charge current flowing, and thus probably also no charge accumulation?

As the referee points out, there is no net current flow –but there is an FMR-induced charge accumulation and AC charge oscillations (see e.g. ref. 53 in the manuscript). At the interface, charge transfer takes place from the metal to C60 due to the equalisation of the metal-semiconductor levels, and there are also present in the molecules free charges due to impurities/defects. During FMR, the charge may become spin polarised and/or diffuse and accumulate at the interface, which would also change its magnetic and dielectric properties. We attribute the observed optical effects to these magneto-electrical changes.

5) Again on the data interpretation. If the enhanced PL is really due to spin accumulation at the metal-organic interface, this would mean that the whole PL signal comes from the interface. Is this really the case? Is the method sensitive enough to detect such a small signal? vWhat about all the rest of the C60 molecules that are not in contact with Py? How thick are the C60 films?

Given that the photoluminescence of C60 on Py is about one order of magnitude lower than for C60 on an insulator, the observed factor 2 change during spin pumping does imply that it is the molecular layers closer to the interface that are most affected. If the spin pumping leads to a full cancellation of the dipole energy transfer effect, some ~5 molecular layers would be involved in the measured change (1/5 of the film thickness). However, it is more likely that the effect is at its maximum at the interface and then decays away from it (perhaps following the exponential decay of the spin current propagation), rather than there being a sharp boundary. The effect of exciton quenching in close proximity to metal surfaces for molecular layers of up to 20 nm thick can be seen e.g. in ref. 41. We have made changes in the manuscript to reflect this discussion.

6) Sample preparation: how did you check that C60 are still intact after Py is deposited on top? How about intercalation of Py inside the C60 films? How well defined are the interfaces in the used devices?

This is indeed an important point that we have extensively considered. We have done cross-sectional TEM of metals sputtered on C60 (see for example our previous work in T. Moorsom et al. PRB 2014 and F. Al Ma'Mari et al. Nature 2015). We have also done extensive reflectivity measurements (X-rays and neutrons) in our multilayers that ensure that any interdiffusion extends at most for one or two molecular layers (chemical roughness obtained at the interface). This is in agreement with the results found in other groups, which observed a ~ 1 nm interdiffusion region for sputtered magnetic films on C60. Differently from other 'soft' molecules, C60 has a crystal structure at room temperature and it is not easily permeated by metal ions –one of the reasons we chose this compound, in addition to its electro-optical properties.

Reviewer #2 (Remarks to the Author):

This paper reports modulation of optical properties of C₆₀ (fullerene) put on a ferromagnet (Py) induced by microwave irradiation. They observed modulation of optical absorbance, vibration Raman scattering peaks (shape, position, and intensity), and photo luminescence intensity.

Spin current experiments have relied on the inverse spin Hall effect for long years, and discovery of another new method will become an important piece of information for spintronics researchers.

However, the paper does not show that the observed signals are relevant to spin current, although the title is "Optical conversion of pure spin current..." The data just shows the spectral modulation at the FMR condition. No typical spin-related phenomenon, such as circular dichroism, polarization rotation, oscillatory angular dependence, or comparison with theories is shown. Indeed, one of the effects of FMR is spin pumping, but it is not all of FMR; there are some other effects, such as local heating, mechanical oscillation etc. Local inhomogeneous heating may induce strain, for instance, which cannot be blocked by Al₂O₃.

Therefore, I regret to conclude that the "spin current induced" is just speculation, and the paper does not meet the criteria of Nature communication.

We thank the referee for her/his comments. Following the suggestions above, we altered our experimental setups to perform new measurements on light irradiation, light polarisation and the correlation with the magnetic field/spin current.

Firstly, we have measured the effect of light irradiation and polarisation in the spin mixing conductance in a range of samples, please see the text in page 4 and measurements in Figure 1d, showing a increased spin mixing conductance under irradiation and for light polarised parallel to the spin quantization axis (no significant change for light perpendicular to the quantization axis).

We have also measured the changes in light absorbance as a function of the relative direction of the magnetic field with respect to the light polarisation, finding again an enhanced absorbance for light unpolarised or polarised parallel to the field, but reduced otherwise.

In order to consider non-thermal vibrational effects, we have also done measurements using ferromagnets with a larger magnetoelastic coupling (which is extremely small in Py), and we have observed a reduced effect -see the supplementary information. We need to note as well that effects such as heating lead to the opposite of what we observe; i.e. an increase in temperature reduces the optical absorbance, spin mixing conductance and luminescence as well as giving raise to less defined Raman peaks. This is corroborated in our measurements (see SI) and in many other works before.

Finally, our experimental work aims to show for the first time that there is a correlation between spin pumping and the optical properties of molecular layers adjacent to a ferromagnet. Differently from other optical experiments, such as the photocurrents generated in topological insulators, the coupling at the molecular interface is unlikely to be a single and direct mechanism (e.g. through the spin orbit interaction), but more possibly mediated by other effects due to the spin current, and the theory requires combining spintronics, photonics and molecular electronics. We believe our far-reaching experimental results and initial reasoning, together with other pioneering results in close magneto-optic topics such as those presented in refs. 13, 17 etc., will demonstrate the high potential for interesting physics and applications, leading to more efforts in the field, and in particular in the theory behind the magneto-optic coupling in molecular spintronics.

(minor points)

(a) Figure 2 b caption: What does the function $(t^2+x^2)^{-3/2}$ mean? Is it based on a physical model?

Firstly, our apologies -there was a typo in this equation and it should have been $(t^2+x^2)^{-1/2}$. It is an empirical equation related to the magnetic field generated by a finite U-shaped wire, which is dependent of the inverse of the square root of the sum of the square of the distance to the wire and the square of the size of the loop. We use this as a simple model of the field generated close to our relatively broad waveguide.

(b) Page 9 line 9: However, the energy transfer may be screened by the charge and spin accumulation: Please elaborate the mechanism.

We have done so in the new version of the manuscript (see page 13); molecular films on metallic substrates have lower photoluminescence because energy transfer to the metal leads to non-radiate recombination of the excited states. This energy transfer can be screened by a charged and/or polarised interface –which previously has been done by using molecular spacers (see ref. 45). The efficiency of this layer in stopping the energy transfer depends on its electrical permittivity (see eq. 2). An increased permittivity leads to in smaller energy transfer and higher photoluminescence.

Reviewer #1 (Remarks to the Author):

[The referee recommends publication with no further comments for the authors]

Reviewer #2 (Remarks to the Author):

The authors added some data to reinforce his conclusion. I am satisfied with this revision and I can recommend its publication.